# Violence experience by perpetrator and associations with HIV/STI risk and infection: a cross-sectional study among female sex workers in Karnataka, south India

Alicja Beksinska,[1] Ravi Prakash,[2] Shajy Isac,[2] H L Mohan,[2] Lucy Platt,[3] James Blanchard,[4,5] Stephen Moses,[4,5] Tara S Beattie[1]

For numbered affiliations see end of article.

**Correspondence to**
Dr Alicja Beksinska;
A.Beksinska1@uni.bsms.ac.uk

## ABSTRACT

**Objectives** Female sex workers (FSWs) experience violence from a range of perpetrators, but little is known about how violence experience across multiple settings (workplace, community, domestic) impacts on HIV/sexually transmitted infection (STI) risk. We examined whether HIV/STI risk differs by the perpetrator of violence.

**Methods** An Integrated Biological and Behavioural Assessment survey was conducted among random samples of FSWs in two districts (Bangalore and Shimoga) in Karnataka state, south India, in 2011. Physical and sexual violence in the past six months, by workplace (client, police, coworker, pimp) or community (stranger, rowdy, neighbour, auto-driver) perpetrators was assessed, as was physical and sexual intimate partner violence in the past 12 months. Weighted, bivariate and multivariate analyses were used to examine associations between violence by perpetrator and HIV/STI risk.

**Results** 1111 FSWs were included (Bangalore=718, Shimoga=393). Overall, 34.9% reported recent physical and/or sexual violence. Violence was experienced from domestic (27.1%), workplace (11.1%) and community (4.2%) perpetrators, with 6.2% of participants reporting recent violence from both domestic and non-domestic (workplace/community) perpetrators. Adjusted analysis suggests that experience of violence by workplace/community perpetrators is more important in increasing HIV/STI risk during sex work (lower condom use with clients; client or FSW under the influence of alcohol at last sex) than domestic violence. However, women who reported recent violence by domestic and workplace/community perpetrators had the highest odds of high-titre syphilis infection, recent STI symptoms and condom breakage at last sex, and the lowest odds of condom use at last sex with regular clients compared with women who reported violence by domestic or workplace/community perpetrators only.

**Conclusion** HIV/STI risk differs by the perpetrator of violence and is highest among FSWs experiencing violence in the workplace/community and at home. Effective HIV/STI prevention programmes with FSWs need to include violence interventions that address violence across both their personal and working lives.

## Strengths and limitations of this study

► This is the first study to examine the association between violence exposure from multiple perpetrators and HIV/sexually transmitted infection risk and prevalence among female sex workers (FSWs).

► The study used a robust sampling strategy and had a reasonably large sample size (>1000 FSWs).

► Intimate partner violence (IPV) was assessed using the validated eight-item WHO IPV questionnaire.

► However, violence experience by other perpetrators was assessed used a two-item question which may have led to under-reporting of workplace/community violence.

► The categorisation of violence by perpetrators was based on crude definitions, which likely do not reflect the fluidity of relationships (eg, client to intimate partner and vice versa).

► Some associations may have been due to chance, particularly for outcomes with small numbers, such as syphilis infection.

## INTRODUCTION

Violence, in particular, gender-based violence, is recognised as a risk factor for HIV and sexually transmitted infections (STIs).[1] Female sex workers (FSWs) experience high levels of violence and HIV/STIs.[2] Recent estimates indicate that FSWs have a lifetime violence prevalence of 41%–65%[3] compared with 27.8%–32.2%[4] among women in the general population as well as 13.5 (95% CI 10.0 to 18.1) times the odds of HIV infection.[5] FSWs commonly experience violence on entry into sex work when they are at their most vulnerable.[2] FSWs can experience violence in their workplace from a range of perpetrators including police, clients, pimps and madams,[6–10] as well as in their community from private militias, religious groups and others who may perceive sex workers

to be 'immoral' and blame them for the spread of HIV and STIs.[6] FSWs also experience high levels of domestic violence, from intimate partners.[11 12]

Violence against FSWs is associated with increased HIV/STIs[7 8] and STI symptoms,[7 13] and can hinder HIV prevention programming.[11] Recent violence experience may directly increase HIV/STI risk through condom breakage/failure or condom non-use.[14–16] Furthermore, men who perpetrate violence against women are more likely to engage in high-risk behaviours including having multiple sexual partners, high alcohol consumption and inconsistent condom use, and have an increased prevalence of HIV, STIs and STI symptoms. This puts their sexual partners at increased HIV/STI risk.[2] HIV vulnerability may be increased indirectly as fear of police violence or arrest may result in women not carrying condoms or working in more isolated, dangerous locations[17] and deter them from accessing sexual health services.[6] Alcohol use is common among FSW populations[18] and their clients and is associated with increased HIV/STI risk[19] and violence experience.[20 21]

India has the third largest HIV epidemic globally, with prevalence rates among FSWs ranging from 2% to 38%. Karnataka state in south India has one of the highest HIV burdens among FSWs, with prevalence previously reaching >30% in some districts.[7] Although sex work per se is not illegal, many FSWs and police wrongly understood this to be the case and sex work is highly stigmatised.[21 22] Violence against FSWs has been identified as a key concern.[8] In 2003, the Karnataka Health Promotion Trust, in partnership with the University of Manitoba, was established to scale up HIV prevention programming with 'high-risk' populations. At scale, the intervention worked with >60 000 FSWs per annum using a rights-based approach to address violence, stigma and poverty as part of comprehensive HIV prevention programming.[22] Changes in behaviour and HIV and STI prevalence were assessed using serial Integrated Biological and Behavioural Assessment (IBBA) cross-sectional surveys.

Studies examining the association between violence and HIV/STI infection and sexual risk behaviours among FSWs have primarily focused on client violence. Although there is now evidence of how HIV prevention programmes among FSWs can effectively reduce violence from non-partners,[7 8 21] there has been less research on the impacts of domestic violence on HIV/STI risk or the efficacy of programmes targeting domestic violence among FSWs. The complexity of violence from different perpetrators and the associated HIV/STI risks is still unclear which hinders the ability of researchers and policymakers to design violence prevention programmes. Qualitative research has suggested that domestic violence may be as important as workplace violence in contributing to HIV/STI risk,[23] and FSWs report low levels of condom use with intimate partners.[24] To our knowledge, only one previous study in Andhra Pradesh, India, which examined violence from husbands and clients, found an association between husband-perpetrated violence and increased risk of

inconsistent condom use with clients.[25] However, this study did not examine prevalence of biological outcomes (HIV/STI prevalence) and did not include non-marital intimate partners or other workplace/community perpetrators. FSWs also face violence in their wider community. Previously, violence from 'rowdies' (gang leaders/members) and 'strangers' has been reported in India[7 8] but there is currently no research on how violence in the community impacts HIV/STI risk. Additionally no studies have examined the risks associated with experiencing violence from multiple perpetrators, that is, from domestic and non-domestic (workplace/community) perpetrators. As a result, there is a need to better understand how violence from different and/or multiple perpetrators impacts on HIV/STI infection and sexual risk behaviours among FSWs.

This study aims to address this gap in the current literature by describing the distribution of workplace, community and domestic perpetrators of violence among FSWs in Karnataka and examining whether HIV/STI infection and sexual risk behaviours differ depending on the perpetrator of violence.

## METHODS

### Study design

Data were collected from two districts (Shimoga and Bangalore) in the third round of a series of IBBA surveys, in Karnataka. Intervention programmes were first implemented in 2004. Round 3 IBBA surveys took place in July and August 2011.[7]

Sample size calculations have been reported previously.[8] In brief, the target sample for each IBBA district was fixed at 400. To represent the greater number of FSWs in Bangalore and the variation in sex work typology, a sample size of 800 was used.[8 26] Following mapping of FSWs across the two districts, two sampling methods were used. For FSWs working at brothels, lodges, homes and *dhabas* (road-side eating establishments) with a more fixed population, a conventional cluster sampling method was used. For street-based FSWs, time-location cluster sampling was used. Inclusion criteria were women aged 18–49 years who had received money or gifts in exchange for sex at least once in the past month. FSWs gave written or witnessed verbal informed consent and were interviewed by trained female interviewers in a rented room close to their workplace.[8 26 27] No identifying information was recorded.

The behavioural questionnaire was prepared in English and then translated into the local language, Kannada. It included one question on non-partner physical violence (*'In the last six months, how many times would you say someone has beaten you? (hurt, hit, slapped, pushed, kicked, punched, choked, burned?) Who did this to you?'*) and one question on non-partner sexual violence (*'in the past one year, has anyone besides your main partner ever forced you to have sexual intercourse when you did not want to? If yes, who was/were this/these person/s?'*).[7] Women were given a list of perpetrators to select from as well as the option to qualitatively

report 'other' perpetrators. In round 3 in Bangalore and Shimoga, detailed questions on physical (six items) and sexual violence from non-paying intimate partners (two items) in the last 12 months were also included based on WHO operational definitions of violence[28] (online supplementary appendix A). Due to the two different timeframes (6 and 12 months), the term 'recent' violence will refer to the past 6/12 months.

## Laboratory methods

Blood samples were taken to test for HIV and syphilis. A confirmed syphilis infection was defined by having a Rapid Plasma Reagin (RPR) positive and a Treponema Pallidum Haemagglutination Assay positive with an RPR titre of >1:8 classified as high-titre syphilis; high-titre syphilis is indicative of recent syphilis infection. Further details of laboratory methods have been previously reported.[27]

## Statistical analyses

The analysis was carried out in STATA V.13.1. To take account of sampling probabilities at district, primary sampling unit and individual levels, as well as rates of non-response, data were appropriately weighted. The main exposure, violence, was categorised into workplace perpetrators (clients, police, pimps, madams and coworkers); community perpetrators (strangers, rowdies, neighbours, auto drivers, assistant ward boys, friends and relatives); and domestic perpetrators (husbands, regular partners and lovers). This classification was based on assumptions about which environment (domestic, workplace or community) violence is most likely to have been perpetrated in. In our preliminary analysis, we examined community and workplace violence separately but found the results were very similar; due to the small number of community perpetrators, we decided to collapse this into one category, to create four categories of exposure: 'no violence', 'domestic violence only', 'workplace and/or community violence only' and 'domestic and workplace/community violence'. The primary outcomes were HIV, syphilis and STI symptom prevalence (STI symptoms were self-reported vaginal discharge, lower abdominal pain not associated with menses and/or genital ulcer in the past 12 months). Secondary outcomes included condom use at last sex; condom breakage at last sex; client or FSW under the influence of alcohol during last sex; STI clinic visit in the past six months; and contact with a peer educator in the past month. Associations were measured using ORs, and p values were obtained using the Wald $\chi^2$ test. As the data were weighted and analysed using survey set commands, we used a joint hypothesis test, the adjusted Wald test, to obtain p values using testparm in Stata. This tests the null hypothesis that the coefficients are simultaneously equal to zero, and therefore tests whether there is variation between categories of exposure to violence. For multivariate analysis, age and district were selected as a priori confounders. Confounders were identified separately for each outcome using a change-in-estimate approach, but to increase the uniformity of the

multivariate models, all outcomes were finally adjusted for the same variables. We did not adjust each outcome for all the other outcomes due to co-linearity between many of the main outcomes. The adjusted Wald test was used to test for effect modification.

## Patient and public involvement

FSW community-based organisations (CBOs), implementing partners and FSW peer educators were involved in the design of the questionnaire and recruitment of women. The results were disseminated back to the community via presentations to the CBOs and the implementing partners.

## RESULTS

### Study population and violence experience

Overall, 1111 FSWs participated in the study (Shimoga (n=393), Bangalore (n=718)). Over one-third (34.9%) of FSWs reported recent (past 6/12 months) physical and/or sexual violence with recent physical violence (29.6%) more prevalent than recent sexual violence (21.9%) (table 1). Reported domestic violence experience was high, with 60% of FSWs reporting intimate partner violence (IPV) in their lifetime and over a quarter of women (27.1%) reporting recent domestic violence (past 12 months). Recent workplace violence (past six months) was reported by 11.1% of FSWs, with sexual violence (8.2%) more prevalent than physical violence (5.4%). Workplace violence was mainly perpetrated by clients (9.2%), with <1% perpetrated by police, coworkers and pimps. Recent violence by perpetrators from the community (past six months) was the least prevalent (4.1%) and was perpetrated mainly by strangers (2.1%) and 'rowdies' (1.1%) (table 1).

The venn diagram in figure 1 shows the proportion of women experiencing violence from different perpetrators, and the overlap between violence experienced by workplace, community and domestic perpetrators. Thus, of the 34.9% of FSWs who reported recent violence, 6.8% reported violence by two or more different perpetrator types, and 6.2% reported violence by domestic and workplace or community perpetrators.

The mean age of respondents was 32.9 years, and 54.5% were illiterate (table 2). Two-thirds (66.2%) had a regular partner, and the majority of women had at least one child. Two-thirds (66.1%) had an additional income to sex work. Women solicited clients either by phone (56.7%) or from public places (32.5%). The median number of clients entertained per week was 6 (range 1–70; IQR 4–10) and 15.6% had ever practised sex work outside the district.

Due to the small number of women who reported community violence, for the remaining analyses, workplace and community violence were combined into one category 'workplace/community violence'. Among FSWs who experienced recent violence, socio-demographic and sex work characteristics differed by the perpetrator of violence (table 2). Women who reported recent

**Table 1** Physical and sexual violence by perpetrator

| Type of violence, by perpetrator | | Recent physical violence (%) | Recent sexual violence* (%) | Recent physical and/or sexual violence (%) |
|---|---|---|---|---|
| Overall | | 29.6 | 21.9 | 34.9 |
| Recent domestic violence | | 25.1 | 14.7 | 27.1 |
| | Husband/regular partner | 25.1 | 14.7 | 27.1 |
| Recent workplace violence | | 5.4 | 8.2 | 11.1 |
| | Client | 4.0 | 7.2 | 9.2 |
| | Police | 0.5 | 0.9 | 0.9 |
| | Co-worker | 1.0 | 0.0 | 1.0 |
| | Pimp | 0.0 | 0.2 | 0.2 |
| Recent community violence | | 2.7 | 2.9 | 4.2 |
| | Strangers | 1.6 | 1.1 | 2.1 |
| | Rowdies† | 0.7 | 1.0 | 1.1 |
| | Neighbours | 0.3 | 0.0 | 0.3 |
| | Auto driver | 0.1 | 0.0 | 0.1 |
| | Assistant ward boy‡ | 0.0 | 0.05 | 0.1 |
| | Relatives | 0.2 | 0.4 | 0.5 |
| | Friends | 0.0 | 0.4 | 0.4 |

Missing observations:

*n=19 (1.7%).

†Rowdies: a member or leader of a gang, who has committed offences punishable under the Indian Penal Code.

‡Assistant ward boy: healthcare worker.

workplace/community violence were more likely to solicit clients from public places (53.7%), whereas women who reported recent domestic violence only were more likely to solicit clients by phone (53.9%). A higher median number of clients per week was reported among women who experienced workplace/community violence (9; IQR 5–12) or violence by both domestic and workplace/

**Figure 1** Proportional venn diagram showing overlapping of physical and/or sexual violence experiences among female sex workers by perpetrator.

community perpetrators (9; IQR 6–15), and these women were more likely to have migrated for sex work compared with women who had experienced domestic violence only or no violence (table 2). Women who reported recent violence by both domestic and workplace/community perpetrators had the lowest mean age at start of sex work (25.4 years) and lowest mean age at first sex (15.4 years).

### HIV/STI risk

Overall HIV prevalence was 8.2%, reactive syphilis 3.1% and high-titre syphilis 0.5%. In multivariate analysis, there was no evidence of an association between violence by perpetrator and either HIV (p value 0.27) or reactive syphilis (p value: 0.76) (table 3). However, there was strong evidence (p value<0.0001) for an increased odds of high-titre syphilis infection among women who reported recent violence by both domestic and workplace/community perpetrators compared with women who reported no recent violence (adjusted OR (aOR): 24.96; 95% CI 5.94 to 96.70).

Self-report of STI symptoms (vaginal discharge/genital ulcers/abdominal pain not associated with menses) in the past year was higher among women who reported recent violence compared with FSWs who reported no violence. In multivariate analyses, there was strong evidence for an increased odds of STI symptoms in all categories of violence by perpetrator, with those who experienced violence by both domestic and workplace/community perpetrators having the highest odds of STI symptoms (aOR 3.90; 95% CI 2.10 to 7.26) (table 3).

**Table 2** Socio-demographic and sex work characteristics of female sex workers in Shimoga and Bangalore and associations with violence by perpetrator

| | | | Recent violence by perpetrator | | | | |
|---|---|---|---|---|---|---|---|
| **Characteristic** | | **Overall** | **No violence** %(n=727) | **Domestic violence only** %(n=216) | **Workplace and/or community violence only** %(n=80) | **Domestic and workplace/ community violence** %(n=69) | **P values (χ² test)** |
| Age (years) | <25 | 13.1 | 12.2 | 13.3 | 22.0 | 12.7 | 0.18 |
| | 25–29 | 22.4 | 20.9 | 24.6 | 21.9 | 36.1 | |
| | 30–39 | 45.2 | 44.6 | 47.4 | 41.0 | 42.7 | |
| | 40+ | 19.3 | 22.4 | 14.6 | 15.1 | 8.5 | |
| | Mean | 32.9 | 33.4 | 32.2 | 31.1 | 30.8 | |
| Literacy | Illiterate | 54.5 | 56.2 | 56.2 | 37.8 | 57.9 | 0.07 |
| Marital status | Lives alone | 44.2 | 53.3 | 8.3 | 78.7 | 18.1 | <0.0001 |
| | Lives with partner other than husband | 4.5 | 4.4 | 4.6 | 5.8 | 4.6 | |
| | Married and lives with husband | 51.2 | 42.3 | 87.0 | 15.5 | 76.2 | |
| Regular partner | Yes | 66.2 | 58.3 | 95.1 | 46.6 | 88.2 | <0.0001 |
| Number of children | 0 | 9.7 | 8.7 | 7.3 | 22.5 | 12.2 | 0.03 |
| | 1–2 | 60.3 | 62.0 | 60.1 | 57.8 | 53.9 | |
| | 3+ | 30.1 | 29.4 | 32.6 | 19.7 | 34.0 | |
| | Mean | 2.0 | 2.0 | 2.2 | 1.7 | 2.0 | |
| District | Bangalore | 50.8 | 54.0 | 42.1 | 64.1 | 50.2 | 0.022 |
| | Shimoga | 49.3 | 46.0 | 57.9 | 35.9 | 49.8 | |
| Additional income to sex work* | Yes | 66.1 | 67.5 | 67.7 | 58.5 | 59.5 | 0.31 |
| Age at first sex (years) | <15 | 48.2 | 49.4 | 44.9 | 38.9 | 64.6 | 0.07 |
| | 15+ | 51.8 | 50.6 | 55.1 | 61.1 | 35.4 | |
| | Mean | 16.1 | 16.1 | 16.3 | 16.6 | 15.4 | |
| Age started sex work (years) | <20 | 5.5 | 5.6 | 4.8 | 8.2 | 5.7 | 0.056 |
| | 20–24 | 25.2 | 22.5 | 26.8 | 31.1 | 37.6 | |
| | 25–29 | 29.0 | 26.8 | 33.5 | 30.9 | 32.5 | |
| | 30+ | 40.3 | 45.1 | 35.0 | 29.8 | 24.3 | |
| | Mean | 28.3 | 28.9 | 27.7 | 26.9 | 25.4 | |
| Place of solicitation of sex work | Home | 7.4 | 8.4 | 7.9 | 3.1 | 4.7 | 0.0008 |
| | Rented room/lodge/brothel | 3.4 | 2.6 | 6.7 | 0.6 | 2.4 | |
| | Public place/tamasha/other | 32.5 | 28.3 | 31.5 | 53.7 | 43.4 | |
| | Phone | 56.7 | 60.7 | 53.9 | 42.6 | 49.4 | |
| How much charged for sex with last client (rupees) | 400+ | 53.1 | 53.1 | 52.2 | 62.6 | 46.7 | 0.43 |
| | Mean | 459.3 | 469.8 | 442.5 | 458.5 | 422.3 | |
| Number of clients/week | 1–4 | 28.4 | 28.2 | 34.4 | 18.2 | 12.8 | <0.0001 |
| | 5–9 | 45.0 | 46.6 | 47.3 | 37.7 | 42.0 | |
| | 10+ | 26.6 | 25.2 | 18.3 | 44.1 | 45.2 | |
| | Median | 6.0 | 6.0 | 6.0 | 9.0 | 9.0 | |
| Migrant sex work (ever practiced sex work outside the district and/ or in Mumbai) | Yes | 15.6 | 12.6 | 11.9 | 39.9 | 25.8 | <0.0001 |

Missing observations:
*n=6 (0.5%).
†n=1 (0.1%).

**Table 3** Violence by perpetrator and associations with HIV/sexually transmitted infection (STI) prevalence and sexual risk behaviours

| | | Recent violence from any perpetrator | | | Violence by perpetrator (reference group: no recent violence) | | | |
| --- | --- | --- | --- | --- | --- | --- | --- | --- |
| | | No recent violence (%) (n=727) | Any recent violence (%) (n=365) | P values | Domestic violence only (%) (n=216) | Workplace and/or community violence only (%) (n=80) | Domestic and workplace or community violence (%) (n=69) | P values* |
| HIV | % | 8.1 | 6.1 | | 2.5 | 13.4 | 8.9 | |
| | Crude OR | 1.0 | 0.73 (0.42–1.27) | 0.26 | 0.28 (0.11–0.73) | 1.75 (0.88–3.5.0) | 1.11 (0.40–3.38) | 0.022 |
| | Adjusted OR | 1.0 | 0.82 (0.44–1.53) | 0.53 | 0.40 (0.15–1.09) | 1.16 (0.55–2.44) | 1.32 (0.41–4.29) | 0.27 |
| Reactive syphilis† | % | 3.4 | 2.9 | | 1.5 | 6.2 | 3.5 | |
| | Crude OR | 1.0 | 0.87 (0.40–1.91) | 0.74 | 0.42 (0.09–2.02) | 2.04 (0.60–6.89) | 1.12 (0.30–4.22) | 0.60 |
| | Adjusted OR | 1.0 | 1.27 (0.68–2.38) | 0.46 | 1.14 (0.30–4.46) | 1.17 (0.57–2.40) | 2.04 (0.53–7.81) | 0.76 |
| High-titre syphilis (recent syphilis) | % | 0.38 | 0.64 | | 0 | 0.9 | 2.5 | |
| | Crude OR | 1.0 | 1.70 (0.36–8.03) | 0.50 | – | 2.36 (0.25–22.07) | 6.74 (1.15–39.58) | 0.11 |
| | Adjusted OR | 1.0 | 2.22 (0.54–9.17) | 0.27 | – | 2.27 (0.26–19.8) | 24.96 (5.94–96.70) | <0.0001 |
| STI symptoms in the past 12 months (vaginal discharge, lower abdominal pain not associated with menses and/or genital ulcer) | % | 30.7 | 48.9 | | 41.5 | 57.3 | 63.3 | |
| | Crude OR | 1.0 | 2.16 (1.61–2.89) | <0.0001 | 1.60 (1.11–2.31) | 3.03 (1.77–5.18) | 3.90 (2.18–6.95) | <0.0001 |
| | Adjusted OR | 1.0 | 2.27 (1.66–3.09) | <0.0001 | 1.87 (1.24–2.81) | 2.41 (1.40–4.17) | 3.90 (2.10–7.26) | <0.0001 |
| Condom use last sex with occasional client‡ | % | 97.5 | 94.5 | | 97.2 | 91.6 | 91.0 | |
| | Crude OR | 1.0 | 0.45 (0.21–0.96) | 0.038 | 0.87 (0.29–2.63) | 0.28 (0.10–0.74) | 0.26 (0.09–0.75) | 0.0073 |
| | Adjusted OR | 1.0 | 0.39 (0.19–0.83) | 0.014 | 1.03 (0.33–3.28) | 0.20 (0.07–0.52) | 0.22 (0.06–0.81) | 0.0001 |
| Condom use last sex with regular client§ | % | 93.0 | 88.2 | | 92.6 | 85.9 | 76.0 | |
| | Crude OR | 1.0 | 0.56 (0.32–0.98) | 0.043 | 0.94 (0.44–2.01) | 0.46 (0.20–1.05) | 0.24 (0.11–0.50) | 0.0012 |
| | Adjusted OR | 1.0 | 0.61 (0.32–1.15) | 0.12 | 1.25 (0.54–2.90) | 0.33 (0.15–0.73) | 0.25 (0.10–0.59) | 0.0003 |
| Condom use at last sex with regular partner | % | 23.1 | 13.8 | | 12.2 | 27.8 | 10.4 | |
| | Crude OR | 1.0 | 0.53 (0.32–0.89) | 0.016 | 0.46 (0.26–0.84) | 1.29 (0.51–3.22) | 0.39 (0.16–0.93) | 0.012 |
| | Adjusted OR | 1.0 | 0.63 (0.35–1.14) | 0.13 | 0.79 (0.42–1.51) | 0.40 (0.14–1.12) | 0.48 (0.14–1.67) | 0.30 |
| Condom breakage at last sex¶ | % | 1.2 | 5.1 | | 3.0 | 3.1 | 15.1 | |
| | Crude OR | 1.0 | 4.38 (1.91–10.02) | 0.0005 | 2.46 (0.84–7.25) | 2.60 (0.62–10.9) | 14.3 (5.10–40.30) | <0.0001 |
| | Adjusted OR | 1.0 | 4.32 (1.74–10.73) | 0.0017 | 3.72 (1.13–12.25) | 1.71 (0.36–8.20) | 19.29 (5.42–68.73) | 0.0001 |
| Either client, female sex workers or both under the influence of alcohol at last sex | % | 35.8 | 42.4 | | 34.1 | 53.9 | 56.0 | |
| | Crude OR | 1.0 | 1.32 (0.99–1.76) | 0.058 | 0.93 (0.66–1.30) | 2.09 (1.24–3.52) | 2.28 (1.31–3.99) | 0.0015 |
| | Adjusted OR | 1.0 | 1.29 (0.09–1.77) | 0.12 | 0.97 (0.66–1.42) | 1.66 (0.96–2.84) | 2.16 (1.19–3.92) | 0.024 |

Continued

**Table 3** Continued

| | | Recent violence from any perpetrator | | | Violence by perpetrator (reference group: no recent violence) | | | |
| --- | --- | --- | --- | --- | --- | --- | --- | --- |
| | | No recent violence (%) (n=727) | Any recent violence (%) (n=365) | P values | Domestic violence only (%) (n=216) | Workplace and/or community violence only (%) (n=80) | Domestic and workplace or community violence (%) (n=69) | P values* |
| Visited an STI clinic in the past six months for STI symptoms** | % | 27.8 | 44.5 | | 39.0 | 51.6 | 53.2 | |
| | Crude OR | 1.0 | 2.08 (1.49–2.92) | <0.0001 | 1.66 (1.10–2.52) | 2.77 (1.61–4.78) | 2.95 (1.61–5.43) | <0.0001 |
| | Adjusted OR | 1.0 | 2.28 (1.59–3.27) | <0.0001 | 2.04 (1.28–3.24) | 2.32 (1.35–3.97) | 3.18 (1.68–6.03) | 0.0001 |
| Had contact with a peer educator in the last month†† | % | 92.0 | 96.9 | | 96.3 | 100 | 95.2 | |
| | Crude OR | 1.0 | 2.74 (1.24–6.07) | 0.013 | 2.27 (0.86–6.00) | – | 1.73 (0.55–5.43) | 0.20 |
| | Adjusted OR | 1.0 | 2.22 (0.98–5.00) | 0.055 | 1.75 (0.63–4.90) | – | 1.18 (0.36–3.92) | 0.56 |

Models adjusted for age, district, marital status, migrant sex work, place of selling sex and having an income other than sex work.
*Adjusted Wald test: tests the null hypothesis that the coefficients (categories of exposure to violence) are equal to zero.
Missing observations:
†n=1 (0.1%).
‡n=1 (0.1%).
§n=1 (0.1%).
¶n=3 (0.3%).
**n=159 (14.3%).
††n=22 (2.0%).

## Condom use

Recent violence by a specific perpetrator was associated with reduced condom use in that setting (table 3). In adjusted analyses, any recent violence experience, regardless of the perpetrator, was associated with a significant reduction in reported condom use at last sex with occasional and regular clients. In multivariate analysis, recent violence experience by workplace perpetrators, or by domestic and workplace/community perpetrators, was significantly associated with reduced condom use with last occasional client and last regular client, compared with women reporting no recent violence. Overall, just one-fifth (19.5%) of FSWs reported condom use at last sex with a regular partner. Reported condom use with regular partners was lower among women reporting recent domestic violence compared with women reporting no recent domestic violence, although this association did not remain significant in multivariate analyses.

Condom breakage at last sex was more likely among women who reported any recent violence (5.1%) compared with those who did not report recent violence (1.2%). In multivariate analysis, there was strong evidence (p value: 0.0001) for increased condom breakage among women who reported recent domestic violence (aOR 3.72; 95% CI 1.13 to 12.25), with the highest odds among women who reported violence by both domestic and workplace/community perpetrators (aOR 19.29; 95% CI 5.42 to 68.73).

## Alcohol use

In univariate and adjusted analyses, women who reported recent violence by workplace/community perpetrators (53.9%; aOR 1.66; 95% CI 0.96 to 2.84; p value: 0.024) and by both domestic and workplace/community perpetrators (56.0%; aOR 2.16; 95% CI 1.19 to 3.92, p value: 0.024) were more likely to report either themselves, their client or both being under the influence of alcohol at last sex compared with women who reported no violence or domestic violence only.

## Programme exposure

Women who reported any recent violence were more likely to have visited an STI clinic in the last six months (44.5%) compared with those who did not report recent violence (27.8%) with the highest aOR among those who reported recent violence by both domestic and workplace/community perpetrators (aOR 3.18; 95% CI 1.68 to 6.03). Women who had experienced any recent violence (96.9%) were more likely to have had contact with a peer educator in the past month compared with women who had not experienced recent violence (92.0%), with some evidence for this association in multivariate analyses (aOR 2.22; 95% CI 0.98 to 5.00; p value: 0.055).

## DISCUSSION

This is the first study globally, to our knowledge, to examine violence experience among FSWs by perpetrators in the workplace, community and home, and associations with biological outcomes and HIV/STI risk behaviours. We found a high prevalence of violence from a range of perpetrators experienced by FSWs in this setting in India, with recent domestic violence more commonly reported than violence by workplace or community perpetrators. Additionally, we found that HIV/STI risk differed by perpetrator of violence and was highest among women who reported recent violence from multiple perpetrators; women reporting violence by domestic and workplace/community perpetrators compared with women reporting violence from domestic perpetrators only and workplace/community perpetrators only, were significantly more likely to have high-titre syphilis infection and had the highest odds of recent STI symptoms, condom breakage at last sex, alcohol use at last sex and no condom use at last sex with regular clients. This study is the first of its kind to show that increased STI prevalence and HIV/STI risk among FSWs is associated with experience of violence from multiple perpetrators. It also adds to a growing body of research globally, reporting the burden and range of perpetrators of violence among FSWs.

The pathways between violence exposure and increased HIV/STI risk are complex. Theories of risk pathways include HIV/STI risk associated with forced sex with an HIV-infected partner, women fearing violence if they request condom use, and increased high-risk behaviours as a result of the psychological impact of sexual/physical abuse.[2 29] In South Africa, among women from the general population increasing frequency of violence, for example, reporting many versus one episode of violence, has been associated with increased odds of HIV infection,[30 31] suggesting a 'dose–response' effect between violence and HIV/STI risk. Although our study did not measure frequency of violence, it is possible that women who experience violence from multiple perpetrators experience more violence overall than those who experience violence from domestic or workplace perpetrators only. This may help to explain the increased STI prevalence and sexual risk behaviours among women in our study who reported violence from multiple perpetrators.

Despite the high rates of domestic violence, our study findings suggest that violence by workplace/community perpetrators is more important for increasing sexual risk behaviours overall and during sex work compared with domestic violence. Although a previous study with FSWs in India reported an association between husband-perpetrated violence and reduced condom use with clients,[25] in our study, we found no associations between domestic violence and sexual risk behaviours in the workplace, such as condom use with clients. However, women in our study who reported domestic violence only did have increased odds of STI symptoms and condom breakage at last sex compared with women who did not report any recent violence, suggesting that domestic violence is associated with some level of increased HIV/STI risk.

A recent systematic review of domestic violence among women in India estimated the median prevalence of

lifetime and domestic violence in the past year was 41% and 30%, respectively.[32] In our study, FSWs reported a much higher prevalence of lifetime (60.1%) violence and similar rates of recent domestic violence (27.1%). These high levels of domestic violence need to be addressed to reduce impacts on physical and psychological health.[33] So far, HIV prevention programmes with FSWs have focused mainly on reducing workplace violence[7 8 21] and improving condom use with clients.[26] Although there are examples of successful interventions to reduce domestic violence in women in the general population,[34] the efficacy of such interventions among FSWs is unknown. A cluster randomised controlled trial with FSWs, in Karnataka India, aimed at reducing IPV and improving condom use with their lover/husband is currently being assessed, and is the first of its kind to address domestic violence among FSWs.[35]

Prevalence of recent workplace violence was relatively high (11.1%) despite the success of recent violence interventions in Karnataka,[7] with clients the major perpetrators. Reported community violence was low (4.1%) compared with violence from other perpetrators. However, FSWs may be at greater risk of violence from community perpetrators compared with women in the general population due to stigma and dangerous working environments. An important finding was the strong association between having experienced violence by both domestic and workplace/community perpetrators and increased odds of high-titre syphilis, demonstrating biological evidence of increased STI risk. As high-titre syphilis infection indicates recent infection, the direction of the association is more plausible compared with measures of chronic STI infection (HIV and reactive syphilis). Unfortunately in IBBA R3, FSWs were not tested for other incident STIs, due to budget constraints, although violence has been associated with gonorrhoea in previous IBBAs.[7] Self-reported STI symptoms were strongly associated with violence from all perpetrators with the highest odds among those who reported both domestic and workplace/community violence. Although this may indicate STI infection in some cases, self-reported STI symptoms are not a reliable indicator of biological infection.[36] Vaginal discharge in women in India has been linked to depression and psychosocial stress, which may partly explain this association.[37] The reduced odds of condom use with clients among women who reported violence by workplace/community perpetrators and by both domestic and workplace/community perpetrators, but not domestic violence, indicates that the association between violence and HIV/STI risk may be driven by the environment in which the violence occurs (although this is based on the assumption that different perpetrators are associated with particular environments).

The finding that FSWs who report recent violence have higher STI clinic attendance and recent contact with a peer educator reflects positively on the HIV/STI prevention programme in Karnataka, suggesting recent experience of violence does not hinder women from accessing services. In this study, having experienced workplace/community violence and both domestic and workplace/community violence was associated with alcohol use at last sex. Having experienced violence can lead to increased alcohol consumption as a coping mechanism.[19 21] Alternatively being under the influence of alcohol may increase vulnerability to violence and arrest.[19]

This study had strengths and limitations. Although previous research has examined IPV and workplace violence among FSWs,[25] none has included community violence or examined associations with biological STI infection. Only one previous study in Soweto, South Africa, has reported on the prevalence of violence experience from multiple perpetrators among FSWs,[10] but this study did not examine associations with HIV/STI prevalence or risk behaviours. To our knowledge, our study is the first to demonstrate increased prevalence of STI infection and sexual risk behaviours among FSWs who experience violence from multiple perpetrators compared with FSWs who report either no recent violence or recent violence from domestic or workplace perpetrators only. Although the data were collected in 2011, they remain the most recent data on HIV/STI prevalence and risk behaviours among FSWs in Karnataka. In addition, this is one of the few datasets available globally, which assesses exposure to both workplace and domestic violence among FSWs, as well as biological and behavioural markers of HIV and STI risk and infection. Other important strengths were the robust sampling strategy and the reasonably large sample size. With cross-sectional data, it is not possible to ascertain the direction of association for some outcomes or infer causality. Reporting bias may have contributed to over-reporting of certain outcomes (such as condom use) while more stigmatised and sensitive topics (such as alcohol consumption and violence) may have been under-reported. The categorisation of violence by perpetrators was based on crude definitions, which likely do not reflect the fluidity of relationships and environments in which the violence occurs. For example, women who sell sex at home may experience domestic and workplace violence in one physical environment while the definition between regular client and lover/partner can become blurred, with clients becoming lovers and vice versa. Some associations may have been due to chance, particularly for outcomes with small numbers and wide CIs, such as high-titre syphilis infection. Additionally overlaps in CIs between the exposure categories, indicate there is uncertainty in whether there is a true difference in risk by perpetrator of violence. If the WHO standardised 13-item violence questionnaire, which has been shown to yield higher response rates,[8] had also been used for non-partners, it might have increased reporting of violence from workplace and community perpetrators. There was a discrepancy in the timeframe of between recent non-partner physical violence (past 6 months) and recent non-partner sexual violence/intimate partner violence (past 12 months), which could have led to under-reporting of non-partner physical violence.

Despite these limitations, the findings of this study have important implications for HIV/STI prevention among FSWs. Violence against FSWs across both domestic, workplace and community settings needs to be addressed through integrated, comprehensive HIV programmes to enforce their human right to be able to live and work without fear for their safety.

**Author affiliations**
[1]Department of Global Health and Development, London School of Hygiene and Tropical Medicine, London, UK
[2]Karnataka Health Promotion Trust, Bengaluru, Karnataka, India
[3]Department of Social and Environmental Health, Faculty of Public Health and Policy, London School of Hygiene and Tropical Medicine, London, UK
[4]Department of Community Health Sciences, University of Manitoba, Winnipeg, Manitoba, Canada
[5]Department of Medical Microbiology, University of Manitoba, Winnipeg, Manitoba, Canada

**Acknowledgements** The authors thank the women who participated in this study.

**Contributors** AB conducted the analyses and wrote the first draft of the manuscript. RP supervised the analyses and reviewed the article. RP, SI, HLM, LP, JB and SM contributed to the study design and reviewed the article. TSB conceptualised the study, supervised the analyses and reviewed the article.

**Funding** This study was supported by the India AIDS Initiative (Avahan) of the Bill & Melinda Gates Foundation, grant no. OPP52138. TB, RP, LP and SI were supported by the UK Department for International Development (DFID) as part of STRIVE, a 6-year programme of research and action devoted to tackling the structural drivers of HIV (http://STRIVE.lshtm.ac.uk/). TB is supported by a British Academy Fellowship.

**Disclaimer** The views expressed herein are those of the authors and do not necessarily reflect the official policy or position of the Bill and Melinda Gates Foundation, UK DFID or the British Academy.

**Competing interests** None declared.

**Patient consent** Not required.

**Ethics approval** This study was approved by the ethical review board of St Johns Medical College in Bangalore, India (IRB: 179/2010); the Research Ethics Board at the University of Manitoba, Canada (IRB: H2005:098); and the Research Ethics Committee at the London Schoolof Hygiene and Tropical Medicine (IRB: 11118).

**Provenance and peer review** Not commissioned; externally peer reviewed.

**Data sharing statement** No additional data are available.

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
