## [Reviewer comments · BMJ Open]

ARTICLE DETAILS

TITLE (PROVISIONAL)	Violence experience by perpetrator and associations with HIV/STI risk and infection: a cross-sectional study among female sex workers in Karnataka, south India
AUTHORS	Beksinska, Alicja; Prakash, Ravi; Isac, Shajy; Mohan, H L; Platt, Lucy; Blanchard, James; Moses, Stephen; Beattie, Tara

VERSION 1 – REVIEW

REVIEWER	Amee Schwitters CDC, Lesotho
REVIEW RETURNED	04-Mar-2018

GENERAL COMMENTS	Authors clearly state the problem and present findings from the current research
--

REVIEWER	Tommi Gaines University of California San Diego, USA
REVIEW RETURNED	07-Mar-2018

GENERAL COMMENTS	The purpose of this study was to examine female sex workers (FSWs) experiences with sexual and physical violence by different perpetrators (e.g., clients, intimate partners, police, etc.) and to determine whether HIV/STI risk among FSWs varied according to the perpetrator of violence. Using a large sample of FSWs in India, the study found that HIV/STI risk increased under certain situations, with FSWs more likely to report HIV/STI risk-related behaviors if they experienced community and/or workplace violence compared to those not experiencing any violence in the past 12 months. Although the results are interesting, the main concern is that there is no clear indication of how this study advances our knowledge on the relationship between HIV/STI risk and violence. Rather, the study appears to replicate previous findings, which were cited in the introduction, on the link between experiences of violence and HIV/STI risk among FSWs. And the reason that nothing new can be gained from this study is because community violence was not separated from workplace violence; rather the two were combined into a single indicator. It's impossible to determine whether the association between HIV/STI risk and workplace and/or community violence was being driven by a particular perpetrator of violence, or more specifically by client-perpetrated violence since clients were the second most common perpetrator of violence according to table 1. There was insufficient justification as to why the authors chose to collapse community and workplace violence into a single category. Further, the authors highlight the uniqueness of this study is due to the inclusion of community violence but again, this form of violence was combined with workplace violence making it impossible to
---

determine if there are particular perpetrators of violence that increases a FSWs' HIV/STI risk. For example, do FSWs experiencing violence from perpetrators who are neither clients nor domestic partners, more likely to report HIV/STI risk behaviors compared to FSWs reporting client perpetrated violence or intimate partner violence? Moreover, the authors may want to reconsider how they conceptualize the term "environments" since in it appears to be another way to label perpetrator of violence given that participants were not asked the place in which violent act(s) occurred. Additional aspects that can improve the manuscript are detailed below:

Introduction:

The introduction does not include a brief review of literature addressing the shortcomings of current studies on the perpetrators of violence and what new information is to be gained from this study. Some of this is included in the discussion but this rationale should be stated upfront to provide readers with a clear understanding of how this study can advance knowledge on the relationship between violence and HIV/STI risk among FSWs.

Methods:

The measure on perpetrator of violence was not sufficiently described. The statistical analyses defines three categories of perpetrators (workplace, community and domestic) but it is unclear how these categories were constructed from the questionnaire. Did participants select from a pre-defined list or was this an open-ended question? In addition, clarify the terms "Rowdies" and "Assistant ward boy" as these are not common terms for describing perpetrators of violence.

The main exposure variable is reported as "recent physical and/or sexual violence" in the results section but "recent" was never defined in the methods section. It appears that participants were asked about exposure to violence in the past 12 months (among intimate partners) and past 6 months (among non-intimate partners). It will be helpful to clarify how the term "recent" is being defined to aid in the interpretation of the results.

The rationale for including higher titer as a variable needs to be clarified in the methods section since currently, the justification is only provided in the discussion section.

It doesn't appear the p-value was adjusted for multiple comparison; however such an adjustment should have been made given that the authors are considering 8 outcomes.

Results:

Figure 1 is very difficult to interpret and therefore unclear how this adds additional information to what is already presented under Table 1. The primary issue is that there are multiple proportions presented in the diagram but no labels to define which proportion accompanies a particular part of the Venn diagram. Instead, the authors should consider only including the proportions that correspond to sections where the circles overlap and then describing this relationship in the results section (i.e., XX% of FSWs experienced both domestic and workplace violence in the past 12 months, XX% experienced community violence and workplace violence in the past 12 months, etc).

Discussion

The interpretation of the results focus on a dose-response relationship between violence and HIV/STI risk in that FSWs who reported multiple perpetrators in the past 12 (or 6 months) are more likely to report more behaviors related to HIV/STI risk than women

	reporting fewer to no perpetrators of violence. However, such a relationship was never tested in the regression model. Rather, the only test conducted was a Wald test which assessed whether all of the regression coefficients equaled zero (i.e., whether there is an overall association between violence and HIV/STI risk). In table 3, several of the 95% CIs for OR overlap suggesting that there may not be a difference in HIV/STI risk across the different levels of violence (i.e., domestic, workplace/community or domestic and workplace/community). It's unclear why the authors are making statements such as "we found that HIV/STI risk differed by perpetrator of violence and was highest amongst women who reported recent violence across multiple environments" and "our study findings suggest that workplace/community violence is more important for increasing sexual risk behaviours overall and during sex work, compared with domestic violence" when their analytic approach never tested whether HIV/STI risk differed across different perpetrators of violence (i.e., testing differences in HIV/STI risk between the levels of the categorical variable, violence). Minor There were a few grammatical errors that need to be addressed in the overview and introduction.
--	---

VERSION 1 – AUTHOR RESPONSE

Reviewer 1:

Authors clearly state the problem and present findings from the current research

Response: We thank the reviewer for their comments.

Reviewer 2

The purpose of this study was to examine female sex workers (FSWs) experiences with sexual and physical violence by different perpetrators (e.g., clients, intimate partners, police, etc.) and to determine whether HIV/STI risk among FSWs varied according to the perpetrator of violence. Using a large sample of FSWs in India, the study found that HIV/STI risk increased under certain situations, with FSWs more likely to report HIV/STI risk-related behaviors if they experienced community and/or workplace violence compared to those not experiencing any violence in the past 12 months. Although the results are interesting, the main concern is that there is no clear indication of how this study advances our knowledge on the relationship between HIV/STI risk and violence. Rather, the study appears to replicate previous findings, which were cited in the introduction, on the link between experiences of violence and HIV/STI risk among FSWs.

Response: We thank the reviewer for their comments. This is the first study globally, to our knowledge, to have examined violence experience among FSWs by different perpetrators, and associations with biological outcomes and HIV/STI risk behaviours. This research is important because it holistically captures women's violence experience and demonstrates that experiencing violence by perpetrators in the workplace/community and at home is associated with increased syphilis infection and HIV/STI risk behaviours. We have now expanded the introduction to emphasise the current gap in knowledge, the limitations of previous studies and how our study aims to address this gap. (page 5, lines 3-26).

And the reason that nothing new can be gained from this study is because community violence was not separated from workplace violence; rather the two were combined into a single indicator. It's impossible to determine whether the association between HIV/STI risk and workplace and/or community violence was being driven by a particular perpetrator of violence, or more specifically by client-perpetrated violence since clients were the second most common perpetrator of violence according to table 1. There was insufficient justification as to why the authors chose to collapse community and workplace violence into a single category. Further, the authors highlight the uniqueness of this study is due to the inclusion of community violence but again, this form of violence was combined with workplace violence making it impossible to determine if there are particular perpetrators of violence that increases a FSWs' HIV/STI risk. For example, do FSWs experiencing violence from perpetrators who are neither clients nor domestic partners, more likely to report HIV/STI risk behaviors compared to FSWs reporting client perpetrated violence or intimate partner violence?

Response: In our preliminary analysis we did examine community and workplace violence separately but found the results were very similar. We then decided to collapse this into one category due to the small number of community perpetrators. However, we still think it is important that we were able to describe community perpetrators (as in table 1 and figure 1), which has not been done before in previous studies. We have amended the statistical analysis section of the methods to ensure that the justification for this is more clearly explained (see page 7 line 14-19).

Moreover, the authors may want to reconsider how they conceptualize the term "environments" since in it appears to be another way to label perpetrator of violence given that participants were not asked the place in which violent act(s) occurred.

Response: We agree with the reviewers comment about how we have conceptualised the term 'environment' and we are aware that this is a crude way of categorising violence. In response, we have removed the term environment. and used the term 'workplace perpetrators', 'community perpetrators' and 'domestic perpetrators' throughout the manuscript in order to more accurately reflect how the data was collected. We have added an explanation in the statistical analysis section of the methods about how we decided to classify perpetrators (See page 7, line 8-14).

Additional aspects that can improve the manuscript are detailed below:

Introduction

The introduction does not include a brief review of literature addressing the shortcomings of current studies on the perpetrators of violence and what new information is to be gained from this study. Some of this is included in the discussion but this rationale should be stated upfront to provide readers with a clear understanding of how this study can advance knowledge on the relationship between violence and HIV/STI risk among FSWs.

Response: We agree and have now included a more detailed overview of the literature on the relationship between HIV/STI risk and violence among FSWs in the introduction and how our study addresses gaps in the literature. (page 4, line 3-26)

Methods:

The measure on perpetrator of violence was not sufficiently described. The statistical analyses defines three categories of perpetrators (workplace, community and domestic) but it is unclear how these categories were constructed from the questionnaire. Did participants select from a pre-defined list or was this an open-ended question? In addition, clarify the terms "Rowdies" and "Assistant ward boy" as these are not common terms for describing perpetrators of violence.

Response: Participants were given pre-defined perpetrators to select from, with an option for 'other', which allowed an open-ended response. We classified perpetrators according to the environment in which violence likely occurred, e.g client, pimp and co-worker violence in the workplace, strangers and rowdies in the community and husbands and lovers in a domestic environment. As there have been no previous studies attempting to classify violence by perpetrator in this way, we have had to create these categories based on local knowledge and assumptions about where violence is likely to occur. We acknowledge that these are crude categorisations and there is likely to be cross-over in these environments. We have clarified how the categories of perpetrator of violence were defined in the methods (page 6, line 25; page 7, lines 8-14) and the limitations are more clearly acknowledged in the discussion (page 19 lines 107-112). We have also defined the terms "assistant ward boy" and "rowdies" in table 1 (page 10).

Method cont.

The main exposure variable is reported as "recent physical and/or sexual violence" in the results section but "recent" was never defined in the methods section. It appears that participants were asked about exposure to violence in the past 12 months (among intimate partners) and past 6 months (among non-intimate partners). It will be helpful to clarify how the term "recent" is being defined to aid in the interpretation of the results.

Response: We broadly defined exposure to violence as 'recent' to differentiate it from 'ever', which is cited elsewhere in the literature. Due to the introduction of the WHO violence questionnaire into the 2011 IBBA survey, exposure to IPV and non-partner sexual violence was asked about in the past 12 months while non-partner physical violence was measured in the past 6 months. We have highlighted this difference more clearly in the methods (page 6, line 29) and results (page 9, line 4). We acknowledge that this discrepancy in measurement of the exposure is a limitation (page 19, line 119-123), which we have now included in the discussion.

Method cont.

The rationale for including higher titer as a variable needs to be clarified in the methods section since currently, the justification is only provided in the discussion section.

Response: We have defined high titre syphilis as meaning recent syphilis infection and have now detailed this in the methods, as suggested. (page 7, line 2)

Method cont.

It doesn't appear the p-value was adjusted for multiple comparison; however such an adjustment should have been made given that the authors are considering 8 outcomes.

Response: In our multivariate analyses we did adjust our results for confounders but we did not adjust each outcome for all the other outcomes. This was not done due to co-linearity between many of the main outcomes. All outcomes were adjusted for the same variables to increase the uniformity of the multivariate models. We have amended the statistical analysis section of the methods to detail this more clearly (page 7, line 29-33).

Results:

Figure 1 is very difficult to interpret and therefore unclear how this adds additional information to what is already presented under Table 1. The primary issue is that there are multiple proportions presented in the diagram but no labels to define which proportion accompanies a particular part of the Venn diagram. Instead, the authors should consider only including the proportions that correspond to sections where the circles overlap and then describing this relationship in the results section (i.e.,

XX% of FSWs experienced both domestic and workplace violence in the past 12 months, XX% experienced community violence and workplace violence in the past 12 months, etc).

Response: We agree and have amended Figure 1 to make it clearer to interpret. We have described the data shown in this diagram more clearly in the results section, as suggested (page 9, line 15-20).

Discussion

The interpretation of the results focus on a dose-response relationship between violence and HIV/STI risk in that FSWs who reported multiple perpetrators in the past 12 (or 6 months) are more likely to report more behaviors related to HIV/STI risk than women reporting fewer to no perpetrators of violence. However, such a relationship was never tested in the regression model. Rather, the only test conducted was a Wald test which assessed whether all of the regression coefficients equaled zero (i.e., whether there is an overall association between violence and HIV/STI risk).

Response: We thank the reviewer for this comment, but did not intend for the results to be interpreted entirely as a dose-response relationship. We use an example from another study showing increased levels of violence are associated with increased HIV/STI risk and hypothesise that this may be one of the pathways which explains why women who experience violence from multiple perpetrators (i.e. from domestic and non-domestic perpetrators) are at greater HIV/STI risk. We have amended the discussion to make this clearer and referred to 2 other studies describing other possible pathways, which may help to explain this association. (page 16, line 17-29)

Discussion cont.

In table 3, several of the 95% CIs for OR overlap suggesting that there may not be a difference in HIV/STI risk across the different levels of violence (i.e., domestic, workplace/community or domestic and workplace/community). It's unclear why the authors are making statements such as "we found that HIV/STI risk differed by perpetrator of violence and was highest amongst women who reported recent violence across multiple environments" and "our study findings suggest that workplace/community violence is more important for increasing sexual risk behaviours overall and during sex work, compared with domestic violence" when their analytic approach never tested whether HIV/STI risk differed across different perpetrators of violence (i.e., testing differences in HIV/STI risk between the levels of the categorical variable, violence).

Response: With regards to the overlap in the 95% CIs, we acknowledge that this suggests there may not be a difference between the categories and this is a limitation of the study. We have revised our conclusions to reflect the level of uncertainty when interpreting these results (page 19, line 114-116).

However, our analytic approach did aim to test whether HIV/STI risk differed between perpetrators of violence. As a result of the data being weighted and analysed using survey set commands, the likelihood ratio test is not valid and therefore we used an adjusted wald test to obtain p-values. The wald test used was a joint hypothesis test obtained using the 'testparm' command in Stata which tests whether 3 or more co-efficients are equal to zero. In other words this tests the null hypothesis that the different categories of violence by perpetrator are simultaneously equal to zero, and therefore tests whether there is evidence for variation between categories of exposure to violence. This approach allows us to answer our question of whether HIV/STI risk differs depending on the perpetrator of violence. In response to this comment, we have clarified and expanded on our statistical analyses in the methods (page 7, line 23-28).

VERSION 2 – REVIEW

REVIEWER	Tommi Gaines
----------	--------------

	University of California San Diego, United States
REVIEW RETURNED	09-May-2018

GENERAL COMMENTS	The authors adequately addressed the concerns raised from the previous review. There are a few minor comments to help clarify sections of the manuscript. On page 4, line 6, clarify whether the numbers in parenthesis reflect the 95% CI: “women in the general population as well as 13.5 (10.0-18.1) times the odds of HIV” On page5, line 21, the word “on” should be removed “...how violence in the community impacts on HIV/STI risk” On page 7, line 20, under methods, explain how STI symptom prevalence was measured. Is this something that participants self-reported during their interviews and for the analysis, did the primary outcome include lifetime or past year or past 6 month STI symptoms? Figure 1 is missing from the resubmission Table 3, the authors should consider adding p-values to test whether the prevalence of the outcomes (e.g., HIV, syphilis, STI symptoms, etc.) differs across perpetrators of violence. This will help strengthen statements in their discussion such as “This may partly explain the increased STI prevalence and sexual risk behaviours amongst women in our study who reported violence from multiple perpetrators.” (lines 27-29 page 16) or “our study findings suggest that violence by workplace/community perpetrators is more important for increasing sexual risk behaviours overall and during sex work, compared with domestic violence” (lines 30-32, page 16) or “The finding that FSWs who report recent violence have higher STI clinic attendance and recent contact with a peer educator reflects positively on the HIV/STI prevention programme in Karnataka” (lines 79-81, page 18). On page 16, lines 9-11 , clarify the comparison group in the sentence “women reporting violence by domestic and workplace/community perpetrators were significantly more likely to have high-titre syphilis infection...(than who?)” On page 16, line 23, can the authors provide examples of what is meant by “highest levels of violence”?
--

VERSION 2 – AUTHOR RESPONSE

Response to reviewers

Reviewer(s)' Comments to Author:

Reviewer: 2

Reviewer Name: Tommi Gaines

Institution and Country: University of California San Diego, United States

Please state any competing interests: None declared

Please leave your comments for the authors below

The authors adequately addressed the concerns raised from the previous review. There are a few minor comments to help clarify sections of the manuscript.

On page 4, line 6, clarify whether the numbers in parenthesis reflect the 95% CI: “women in the general population as well as 13.5 (10.0-18.1) times the odds of HIV”

Response: We have clarified that this reflects a 95% CI (line 6, page 4).

On page 5, line 21, the word “on” should be removed “...how violence in the community impacts on HIV/STI risk”

Response: We have amended the sentence as suggested (line 22, page 5).

On page 7, line 20, under methods, explain how STI symptom prevalence was measured. Is this something that participants self-reported during their interviews and for the analysis, did the primary outcome include lifetime or past year or past 6 month STI symptoms?

Response: STI symptoms were measured as self-reported STI symptoms (vaginal discharge, lower abdominal pain not associated with menses and/or genital ulcer) in the past 12 months. We have now detailed how STI symptom prevalence was measured in the methods (line 20-22, page 7).

Figure 1 is missing from the resubmission

Response: Apologies that Figure 1 was missing. This has now been included.

Table 3, the authors should consider adding p-values to test whether the prevalence of the outcomes (e.g., HIV, syphilis, STI symptoms, etc.) differs across perpetrators of violence. This will help strengthen statements in their discussion such as “This may partly explain the increased STI prevalence and sexual risk behaviours amongst women in our study who reported violence from multiple perpetrators.” (lines 27-29 page 16) or “our study findings suggest that violence by workplace/community perpetrators is more important for increasing sexual risk behaviours overall and during sex work, compared with domestic violence” (lines 30-32, page 16) or “The finding that FSWs who report recent violence have higher STI clinic attendance and recent contact with a peer educator reflects positively on the HIV/STI prevention programme in Karnataka” (lines 79-81, page 18).

Response: We thank the reviewer for this suggestion. We could conduct a wald test to examine different HIV/STI outcomes across different perpetrators by taking no violence as the reference category. However, we are already providing this information with the crude and adjusted ORs and CI in the table. Together with the adjusted wald test provided in the last column we think this provides

sufficient information to evidence the statements in the discussion. Therefore we do not think the addition of p-values for each category of perpetrator of violence will add important information. Additionally as table 3 is already very busy, this may make it more difficult for readers to interpret.

On page 16, lines 9-11, clarify the comparison group in the sentence “women reporting violence by domestic and workplace/community perpetrators were significantly more likely to have high-titre syphilis infection...(than who?)”

Response: We have amended this sentence to clarify the comparison group (lines 9-11, page 16).

On page 16, line 23, can the authors provide examples of what is meant by “highest levels of violence”?

Response: We have detailed an example from one of the papers referenced on the association between increasing frequency of violence and increasing HIV/STI risk (line 23-26, page 16).